# Low Protein Expression of both *ATRX* and *ZNRF3* as Novel Negative Prognostic Markers of Adult Adrenocortical Carcinoma

**DOI:** 10.3390/ijms22031238

**Published:** 2021-01-27

**Authors:** Vania Balderrama Brondani, Amanda Meneses Ferreira Lacombe, Beatriz Marinho de Paula Mariani, Luciana Montenegro, Iberê Cauduro Soares, João Evangelista Bezerra-Neto, Fabio Yoshiaki Tanno, Victor Srougi, José Luis Chambo, Berenice Bilharinho Mendonca, Madson Q. Almeida, Maria Claudia Nogueira Zerbini, Maria Candida Barisson Villares Fragoso

**Affiliations:** 1Unidade de Suprarrenal, Serviço de Endocrinologia e Metabologia, Hospital das Clínicas, Faculdade de Medicina da Universidade de São Paulo, Av. Dr. Enéas de Carvalho Aguiar, 155, São Paulo 05403-900, Brazil; amflacombe@usp.br (A.M.F.L.); beremen@usp.br (B.B.M.); madson.a@hc.fm.usp.br (M.Q.A.); 2Laboratório de Hormônios e Genética Molecular LIM/42, Hospital das Clínicas, Faculdade de Medicina da Universidade de São Paulo, Av. Dr. Enéas de Carvalho Aguiar, 255, São Paulo 05403-900, Brazil; beatrizmariani@usp.br (B.M.d.P.M.); lucianam@usp.br (L.M.); 3Laboratório de Sequenciamento em Larga Escala (SELA), Faculdade de Medicina da Universidade de São Paulo, Av. Dr. Arnaldo, 455, São Paulo 01246-903, Brazil; 4Departamento de Patologia, Faculdade de Medicina da Universidade de São Paulo, Av. Dr. Arnaldo, 455, São Paulo 01246-903, Brazil; ibere.soares@hc.fm.usp.br (I.C.S.); czerbini@usp.br (M.C.N.Z.); 5Serviço de Anatomia Patológica, Instituto do Câncer do Estado de São Paulo (ICESP), Faculdade de Medicina da Universidade de São Paulo, Av. Dr. Arnaldo, 251, São Paulo 01246-000, Brazil; 6Serviço de Oncologia, Instituto do Câncer do Estado de São Paulo (ICESP), Faculdade de Medicina da Universidade de São Paulo, Av. Dr. Arnaldo, 251, São Paulo 01246-000, Brazil; joao.evangelista@hc.fm.usp.br; 7Serviço de Urologia, Hospital das Clínicas, Faculdade de Medicina da Universidade de São Paulo, Av. Dr. Enéas de Carvalho Aguiar, 155, São Paulo 05403-900, Brazil; fabio.tanno@hc.fm.usp.br (F.Y.T.); victor.srougi@usp.br (V.S.); chambojl@uol.com.br (J.L.C.); 8Serviço de Endocrinologia, Instituto do Câncer do Estado de São Paulo (ICESP), Faculdade de Medicina da Universidade de São Paulo, Av. Dr. Arnaldo, 251, São Paulo 01246-000, Brazil

**Keywords:** protein expression, *ATRX*, *ZNRF3*, prognostic factor, adrenocortical carcinoma

## Abstract

Adrenocortical carcinoma (ACC) is a rare malignancy that is associated with a dismal prognosis. Pan-genomic studies have demonstrated the involvement of *ATRX* and *ZNRF3* genes in adrenocortical tumorigenesis. Our aims were to evaluate the protein expression of *ATRX* and *ZNRF3* in a cohort of 82 adults with ACC and to establish their prognostic value. Two pathologists analyzed immuno-stained slides of a tissue microarray. The low protein expression of *ATRX* and *ZNRF3* was associated with a decrease in overall survival (OS) (*p* = 0.045, *p* = 0.012, respectively). The Cox regression for *ATRX* protein expression of >1.5 showed a hazard ratio (HR) for OS of 0.521 (95% CI 0.273–0.997; *p* = 0.049) when compared with ≤1.5; for *ZNRF3* expression >2, the HR for OS was 0.441 (95% CI, 0.229–0.852; *p* = 0.015) when compared with ≤2. High *ATRX* and *ZNRF3* protein expressions were associated with optimistic recurrence-free survival (RFS) (*p* = 0.027 and *p* = 0.005, respectively). The Cox regression of RFS showed an HR of 0.332 (95%CI, 0.111–0.932) for *ATRX* expression >2.7 (*p* = 0.037), and an HR of 0.333 (95%CI, 0.140–0.790) for *ZNRF3* expression >2 (*p* = 0.013). In conclusion, low protein expression of *ATRX* and *ZNRF3* are negative prognostic markers of ACC; however, different cohorts should be evaluated to validate these findings.

## 1. Introduction

Adrenocortical carcinoma (ACC) is a rare and aggressive cancer deriving from the adrenal cortex gland with an incidence of 0.7–2 cases/million/year [1,2,3]. Despite a dismal prognosis, the five-year overall survival (OS) rate varies, ranging from 60 to 80% when the tumor is restricted to the adrenal gland and from 0 to 28% in the advanced stages of the disease [3,4,5,6,7,8]. These data suggest that ACC is a heterogeneous adrenal disorder, presenting different behaviors, probably due to its different biology and genetic signatures [2,8,9].

In previous years, considerable advances in the genetic field of ACC tumorigenesis investigation have occurred, allowing patients to be stratified according to the combination of clinical–hormonal–pathological and genetic alterations.

Prognostic factors have an essential role in determining the aggressiveness of ACC. We can classify tumors according to (A) clinical data (age, hormone-related symptoms, and tumor stage); (B) pathological findings, genetic/protein background, and category of surgical procedure (Weiss system, modified Weiss system, Helsinki score, mitotic count, Ki-67 proliferation marker, SF-1, P53, β-catenin, and resection status); (C) clinical and pathological parameter associations (staging system of European Network for the Study of Adrenal Tumors modified by grade, resection status, age and symptoms—mENSAT plus GRAS); and (D) molecular classifications (chromosomal aberrations, methylome, gene expression alterations, microRNAs dysregulation, and gene mutations) [2,3,4,6,10,11,12,13,14,15,16,17,18,19,20,21,22,23,24,25,26,27,28,29,30,31,32,33,34,35].

Pan genomic studies have identified new genes related to adrenocortical carcinogenesis, providing a refinement of ACC taxonomy, and improving the development of new biomarkers for early diagnosis and staging, conducive to the development of molecularly targeted therapies.

*ATRX* is a chromatin structure regulator and telomere structure maintainer. Somatic pathogenic variants and structural variations, which cause the deletion of multiple exons of *ATRX* were associated with germline *TP53* pathogenic variants in a pediatric cohort [36]. Pinto et al. reported alterations in *TP53* and *ATRX* in 32% of their cohort, and these alterations were associated with advanced disease and poor event-free survival in pediatric patients with adrenocortical tumor [36]. Assie et al. identified somatic *ATRX* alterations in only 4% of their adult ACC cohort. In contrast, *ZNRF3* was the most frequently altered gene, corresponding to 21% of cases [27]. This gene encodes a cell surface transmembrane E3 ubiquitin ligase that acts as a negative feedback regulator of the canonical Wnt signaling (Wnt/β-catenin signaling). Zheng et al. studied an adult ACC cohort and reported a frequency of 19.3% for ACCs with deletions or non-silent somatic pathogenic variants in the *ZNRF3* gene. In comparison, *ATRX* alterations were present in only four cases (4.3%) [31].

The Cancer Genome Atlas (TCGA) ACC Research Network and the European Network for the Study of Adrenal Tumors (ENSAT) including the cohort of Assie et al. cohort [27], are two critical datasets for the study of ACC. TCGA-ACC, from National Cancer Institute, includes a cohort of 184 cases of ACC with a median age 48.5 years (14–83 years), 120 (65%) female, and a frequency of ENSAT staging system as follows: stage 1, 10% (18 of 180 cases); stage 2, 49% (88 cases); stage 3, 21% (38 cases); and stage 4, 20% (36 cases). TCGA-ACC registered 68 (37%) deaths (the data shown in the present study are based upon data generated by TCGA Research Network) [37]. The ENSAT and Assie et al. cohorts include a total of 130 ACC cases, with median age 46 years (18–86 years), 90 (69.2%) female, with a frequency of ENSAT stage as follows: stage 1, 7.2% (9 of 125 cases); stage 2, 53.6% (67 cases); stage 3, 16% (20 cases); and stage 4, 23.2% (29 cases). There were 46 (35.4%) deaths among 130 cases [27]. TCGA-ACC dataset registered the most mutated genes in ACC, including *TP53* (21%), *ZNRF3* (19%), *CDKN2A* (15%), *CTNNB1* (16%), *TERT* (14%), and *PPKAR1A* (11%), while the *ATRX* gene had a frequency of 4.3% of cohort [31]. ENSAT with Assie et al. showed the most frequent mutated genes as *ZNRF3* (21%), *CTNNB1* (16%), *TP53* (16%), and *CDKN2A* (11%). Alterations on *ATRX* were present in 4% of cases [27]. Both cohorts present similar clinical and molecular data.

In this study, we aimed to evaluate the protein expression of the *ATRX* and *ZNRF3* genes in a cohort of 82 adult ACC patients from a unique Complex tertiary center to investigate their potential role as prognostic markers.

## 2. Results

### 2.1. ATRX Protein

Patients with ACC showing high *ATRX* expression, defined as tumor scored 2+ or higher, represented 47.5% of cohort (Figure 1). *ATRX* high expression group had a median age at diagnosis 52.5 years. In contrast, patients with ACC with low *ATRX* expression had a median age of 34.7 years (Z = −3.349; *p* = 0.001; Mann–Whitney U test). *ATRX* was highly expressed in 90% of cases without hypercortisolism at initial presentation, while it was expressed at a low level in 59.2% of cases with hypercortisolism (X^2^(2) = 8.759; *p* = 0.013; Chi-squared test). The group with hypercortisolism presented a median *ATRX* expression score of 0.5 (ranging from 0 to 4), while the non-hypercortisolism group presented a median score of 3.08 (ranging from 0 to 4) (X^2^(2) = 6.728; *p* = 0.035; Kruskal–Wallis test).

The quantitative analysis of *ATRX* expression showed a positive correlation with age at diagnosis (coefficient 0.391; *p* < 0.001; Spearman’s rank correlation coefficient) and negative correlations with tumor size (coefficient -0.263; *p* = 0.02; Spearman’s rank correlation coefficient) and Weiss score (coefficient -0.245; *p* = 0.033; Spearman’s rank correlation coefficient).

Thirty-nine patients from the cohort died due to ACC or its clinical complications during the study period. The *ATRX* expression cut-off point was statistically determined and used for the OS analysis. The cut-off value ≤1.5 for *ATRX* expression had an impact on the Kaplan–Meier survival curve (X^2^(1) = 4.021; *p* = 0.045; Kaplan–Meier method and log rank test) (Figure 2). In addition, the Cox regression of OS showed a hazard ratio (HR) of 0.521 (95%CI, 0.273–0.997) for *ATRX* expression of >1.5 when compared with expression of ≤1.5 (Wald = 3.881; df1; *p* = 0.049; Cox’s proportional hazards model).

### 2.2. ZNRF3 Protein

The high expression of *ZNRF3* protein was defined as the sum of extent and intensity parameters equaling ≥3+ (Figure 1). High *ZNRF3* expression was present in 46.25% of the cohort and was associated with a small tumor size (average size of 9.7 cm), while low expression was associated with a large tumor size (average size of 13.3 cm) at diagnosis (Z = −2.665; *p* = 0.008; Mann–Whitney U test). Tumors with high *ZNRF3* expression presented with lower Weiss score (average 4.7) than the group with low *ZNRF3* expression (average 6.1) (Z = −2.942; *p* = 0.003; Mann–Whitney U test). *ZNRF3* expression was related to the ENSAT staging system. High *ZNRF3* expression was present in almost 89 % of patients with an ENSAT staging system score of 1 at diagnosis, 42.9% with an ENSAT score of 2, 29.4% with an ENSAT score of 3, and 44.4% with an ENSAT score of 4 (*p* = 0.033, Fisher’s exact test). *ZNRF3* protein expression had no relationship with hypercortisolism at clinical presentation (X^2^(2) = 2.035; *p* > 0.05, Chi-squared test). Regarding the outcome analysis, 64.1% of patients with ACC who progressed to death presented with a low level of *ZNRF3* protein expression and had a median survival of 19 months (X^2^(1) = 4.716; *p* = 0.030; Chi-squared test).

The quantitative expression of *ZNRF3* showed negative correlations with tumor weight (coefficient −0.282; *p* = 0.026, Spearman’s rank correlation coefficient), tumor size (coefficient −0.313; *p* = 0.005; Spearman’s rank correlation coefficient), and the Weiss score (coefficient −0.345; *p* = 0.002; Spearman’s rank correlation coefficient). The *ZNRF3* expression cut-off of ≤2 had an impact on the OS analysis ((X^2^(1) = 6.277; *p* = 0.012; Kaplan–Meier method and log rank test) (Figure 3). The Cox regression of OS showed an HR of 0.441 (95% CI, 0.229–0.852) for *ZNRF3* expression >2 compared with the value for an expression level of ≤2 (Wald = 5.942; df1; *p* = 0.015; Cox’s proportional hazards model).

*ZNRF3* is considered a suppressor of the Wnt/β-catenin pathway. Due to this known function, we analyzed the status of β-catenin in the tumoral cells. Active Wnt/β-catenin signaling was characterized by a cytoplasmic/nuclear staining for β-catenin [37]. The analysis demonstrated a positive correlation between *ZNRF3* expression and β-catenin expression in the cell membrane (coefficient 0.202; *p* = 0.004; Spearman’s rank correlation coefficient). These data support the function of *ZNRF3* as a tumor suppressor, keeping β-catenin attached in the cell membrane (Figure 4).

### 2.3. Ki-67 Proliferation Marker and Combined Analysis

The Ki-67 proliferation marker showed positive correlation with tumor weight (coefficient 0.377; *p* = 0.01, Spearman’s rank correlation coefficient), tumor size (coefficient 0.344; *p* = 0.007, Spearman’s rank correlation coefficient), and the Weiss score (coefficient 0.572; *p* < 0.001, Spearman’s rank correlation coefficient). Furthermore, the categorization of Ki-67 into groups (<10%, ≥10% and <20%, ≥20%) impacted the OS analysis (X^2^(2) = 29.363; *p* < 0.001; Kaplan–Meier method and log rank test) (Figure 5). The Cox regression of OS from the analysis of Ki-67 expression showed an HR of 5.802 (95% CI, 2.447–13.726) for the Ki-67 expression grouped in ≥10% and <20% (Wald = 15.925; df1; *p* < 0.001; Cox’s proportional hazards model), and we calculated an HR of 9.041 (95% CI, 3.543–23.073) for the Ki-67 expression grouped in ≥20% (Wald = 21.217; df1; *p* < 0.001; Cox’s proportional hazards model).

Combined analysis was done in groups classified according to the Ki-67 value (>10% or ≤10%) and *ATRX* expression (>1.5 or ≤1.5), and according to the Ki-67 value (>10% or ≤10%) and *ZNRF3* expression (>2 or ≤2). The combination of Ki-67 with *ATRX* and *ZNRF3* did not impact the Kaplan–Meier survival curves (X^2^(3) = 5.206; *p* > 0.05; and X^2^(3) = 6.619; *p* > 0.05, respectively. Kaplan–Meier method and log rank test for both). However, the combination of low *ATRX* expression and low *ZNRF3* expression had an effect on the OS curve (X^2^(3) = 9.867; *p* = 0.02; Kaplan–Meier method and log rank test) (Figure 6). The Cox regression of OS from the combined analysis of *ATRX* and *ZNRF3* expressions showed an HR of 0.314 (95% CI, 0.136–0.725) for the *ATRX* (>1.5) and *ZNRF3* (>2) expression levels (Wald = 7.363; df1; *p* = 0.007; Cox’s proportional hazards model).

### 2.4. Recurrence-Free Survival

The recurrence-free survival (RFS) was defined as the period between the date of complete tumor resection and the date of the first radiological evidence of local or distant recurrence [38]. *ATRX* protein expression was associated with a better RFS when a cut-off >2.7 was statistically established (X^2^(1) = 4.920; *p* = 0.027; Kaplan–Meier method and log rank test) (Figure 7). The Cox regression of RFS showed an HR of 0.332 (95% CI, 0.111–0.932) for *ATRX* expression >2.7 compared with the value for an expression level of ≤2.7 (Wald = 4.365; df1; *p* = 0.037; Cox’s proportional hazards model). The presence of high *ZNRF3* protein expression (score > 2) was positively associated with RFS (X^2^(1) = 6.963; *p* = 0.008; Kaplan–Meier method and log rank test) (Figure 8), which means that the cases with high protein expression presented a greater recurrence-free survival than the group with low protein expression. This corroborates to *ZNRF3* function of tumor suppressor. The Cox regression of RFS showed an HR of 0.333 (95% CI, 0.140–0.790) for *ZNRF3* expression >2 compared with the value for an expression level of ≤2 (Wald = 6.222; df1; *p* = 0.013; Cox’s proportional hazards model). Categorization of the Ki-67 proliferation marker into groups (<10%, ≥10% and <20%, ≥20%) impacted the RFS curve (X^2^(2) = 16.357; *p* < 0.001; Kaplan–Meier method and log rank test), as seen in Figure 9. The Cox regression of RFS from the analysis of Ki-67 expression showed an HR of 5.450 (95% CI, 1.872–15.866) for the Ki-67 expression grouped in ≥10% and <20% (Wald = 9.674; df1; *p* = 0.002; Cox’s proportional hazards model), and we had an HR of 7.691 (95% CI, 2.278–25.963) for the Ki-67 expression grouped in ≥20% (Wald = 10.801; df1; *p* = 0.001; Cox’s proportional hazards model).

### 2.5. Other Variables with Prognostic Value

Overall survival was associated with a tumor weight of > 55 g (X^2^(1) = 13.534; *p* < 0.001; Kaplan–Meier method and log rank test), a tumor size of >7 cm (X^2^(1) = 11.462; *p* = 0.001; Kaplan–Meier method and log rank test), a Weiss score of ≥ 5 (X^2^(1) = 15.231; *p* < 0.001; Kaplan–Meier method and log rank test) and Ki-67 of > 8% (X^2^(1) = 18.294; *p* < 0.001; Kaplan–Meier method and log rank test). The ENSAT staging system (X^2^(3) = 11.799; *p* = 0.008; Kaplan–Meier method and log rank test), the presence of metastasis at diagnosis (X^2^(1) = 9.111; *p* = 0.003; Kaplan–Meier method and log rank test), and the presence of local (X^2^(1) = 6.775; *p* = 0.009; Kaplan–Meier method and log rank test), or distant recurrence (X^2^(1) = 8.688; *p* = 0.003; Kaplan–Meier method and log rank test) also impacted the OS curves. The presence of hypercortisolism at the initial presentation did not impact the OS or RFS curves (X^2^(1) = 3.103; *p* > 0.05; and X^2^(1) = 3.060; *p* > 0.05, respectively. Kaplan–Meier method and log rank test for both).

The Cox regression analysis revealed the following independent variables for death due to ACC: tumor weight (HR 39.052; 95% CI 1.357–1123.525; Wald = 4.572; df1; *p* = 0.032; Cox’s proportional hazards model), tumor size (HR 4.451; 95% CI 1.729–11.458′ Wald = 9.577; df1; *p* = 0.002; Cox’s proportional hazards model), Weiss score (HR 4.040; 95 %CI 1.895–8.611; Wald = 13.073; df1; *p* < 0.001; Cox’s proportional hazards model), Ki-67 proliferation marker (HR 6.360; 95% CI 2.434–16.618; Wald = 14.256; df1; *p* < 0.001; Cox’s proportional hazards model), presence of metastasis at diagnosis (HR 2.971; 95% CI 1.415–6.237; Wald = 8.282; df1; *p* = 0.004; Cox’s proportional hazards model), and disease recurrence (HR 14.985; 95% CI 1.358–165.390; Wald = 4.882; df1; *p* = 0.027; Cox’s proportional hazards model). The principal Cox regression results for OS are presented in the Appendix A.

## 3. Discussion

Malignancies develop protector mechanisms for self-survival. Generally, defects are harbored in evolutionarily conserved pathways, such as the Wnt/β-catenin pathway [39]. However, nine acquired biological capabilities, known as cancer hallmarks, are involved in a multistep tumorigenesis process [40].

Telomeres are responsible for protecting genetic material and avoiding end-to-end fusion due to the repair process [41]. Telomeric shortening results from ineffective replication due to a problem in the DNA replication machinery [41,42,43]. When telomeres reach a critical length, the cell enters a state of senescence, which is considered to be a natural tumor suppressor mechanism in humans [44,45]. Some malignancies present an alternative lengthening of telomeres (ALT) pathway to avoid senescence and apoptosis [41,46]. The *ATRX* gene encodes a chromatin remodeler (*ATRX* protein), which functions in nucleosome stability, DNA replication, transcription, maintenance of telomere and heterochromatin structure and stability [46,47,48,49,50]. Lovejoy et al. suggested that the inactivation of *ATRX* is a significant step in the ALT pathway [51]. Bower et al. showed that the loss of wild-type *ATRX* expression in somatic cell hybrids results in the development of the ALT pathway [52]. Barthel et al. analyzed 9127 patients and 31 cancer types and showed that inactivation of *ATRX* is associated with telomere length elongation, reinforcing the idea of an association between *ATRX* and ALT [53]. *ATRX* is suggested to be a suppressor of the ALT pathway [47]. Cohorts of different malignancies have shown that *ATRX* alterations have prognostic value [54,55,56,57,58,59,60,61,62,63,64,65].

The Wnt/β-catenin pathway is one of the central molecular mechanisms of embryonic development [66]. Although it is essential for the embryonic phase, the Wnt/β-catenin has a role in various malignancies. *ZNRF3* is a transmembrane E3 ubiquitin ligase molecular target of R-spondin that is localized to the plasma membrane [67]. Hao et al. suggested that *ZNRF3* inhibits Wnt/β-catenin signaling by promoting decreases in the membrane levels of frizzled and LRP6 [67]. Basham et al. reported *ZNRF3* homeostatic regulation of the adrenal cortex, showing that its loss leads to severe adrenocortical hyperplasia [68]. In the same way as the *ATRX* gene, multiple cohorts have reported the role of *ZNRF3* in tumorigenesis and its prognostic value in different malignancies [69,70,71,72,73,74,75,76,77].

We presented a younger cohort with median age 38.17 years (range 15.38–85.46 years), with more female patients (63, 76.8%), and more deaths (39, 55%), compared to TCGA-ACC and ENSAT and Assie et al. cohorts [27,37]. However, the frequency of ENSAT system in the cohort was very similar with those of both cohorts. Unfortunately, we did not have access to genetic samples to enable determination of the frequency of alterations in these genes or to calculate correlations between molecular alterations and protein expression levels. Nevertheless, we were able to show the protein expression profiles of *ATRX* and *ZNRF3* in our cohort.

Our objective was to analyze the *ATRX* and *ZNRF3* protein expression in an ACC cohort; we chose the immunohistochemistry technique because of its availability and ease of use, aiming to bring pan-genomic studies closer to the clinical practice. The *ATRX* analysis considered the extent of cells. Unlike Mete et al. [32] who registered only the global loss of *ATRX* expression, we adapted the *ATRX* analysis from other neoplasia studies, intending to be more descriptive [78,79]. The *ZNRF3* evaluation was based on studies from Qiu et al. and Yu et al. [71,73], based their characterization of the extent and intensity of *ZNRF3* cytoplasmic staining.

*ATRX* protein expression is seen in the nuclear compartment of ACC cells (Figure 1). Mete et al. studied *ATRX* protein expression in 43 ACCs in a tissue microarray (TMA) and showed a global loss of nuclear *ATRX* expression in 48% of the cohort [32]. We registered the global loss of *ATRX* expression (score 0) in 47.5% (n = 38) of our cohort. The loss of *ATRX* protein expression and its association with disease-free survival or with the presence of adverse outcomes was not proven in the study by Mete et al. [32]. We demonstrated that low expression of *ATRX*, present in 52.5% of our cohort, impacted negatively on OS and RFS. Moreover, we proved the positive effect of *ATRX* expression (score >1.5) on prognosis with an HR of 0.521 (95% CI, 0.273–0.997) (*p* = 0.049).

Hypercortisolism has a negative impact on the OS and RFS of ACC, acting as a predictor of recurrence and death [26,80,81,82]. Although cortisol-secreting ACCs have been associated with a worse OS [83], this was not shown in our cohort. *ATRX* was highly expressed (median score of 3.08) in the non-hypercortisolism group, and it showed a low expression (median score of 0.5) in the hypercortisolism group. The role of *ATRX* in cortisol secretion is unknown.

The occurrence of Wnt/β-catenin signaling has been established in different malignancies [37,84,85]. *ZNRF3* acts as a tumor suppressor of the Wnt/β-catenin pathway [39,68,69,86,87,88]. Tissier et al. have already demonstrated the role of Wnt/β-catenin in adrenal tumorigenesis [89]. They reported that 77% of an ACC cohort presented with active Wnt/β-catenin signaling through diffuse cytoplasmic/nuclear β-catenin accumulation in tumor cells [89]. We demonstrated that the presence of *ZNRF3* expression is correlated with β-catenin expression in the cell membrane (*p =* 0.004), supporting the idea that *ZNRF3* keeps β-catenin attached in the cell membrane (Figure 4).

The high expression of *ZNRF3* was associated with smaller tumors in our cohort. *ZNRF3* expression was negatively correlated with tumor weight (*p* = 0.026) and tumor size (*p* = 0.005). Qin et al. assessed *ZNRF3* expression in a cohort of gastric cancer patients and showed an association of protein expression with smaller tumor size [70]. They also revealed that *ZNRF3* overexpression causes significantly more apoptosis and lowered proliferation of cancer cells by reducing the level of Lgr5, a component of Wnt/β-catenin signaling, while also reducing Gli1, a component of Hedgehog signaling (SHH) [70]. Gomes et al. showed that the SHH pathway is upregulated in adult ACC and downregulated in pediatric adrenocortical tumors [90]. If *ZNRF3* has a role in the adrenal SHH pathway, it remains unclear.

Qiu et al. demonstrated the downregulation of *ZNRF3* protein expression in human papillary thyroid carcinoma cell lines [73]. They showed that *ZNRF3* overexpression strongly inhibits the migratory and invasive capacities of the cancer cells [73]. Yu et al. revealed that strong expression of *ZNRF3* had a significant association with disease-free survival and OS in a colorectal carcinoma cohort [71]. We showed that low *ZNRF3* expression impacted negatively on OS analysis in this ACC cohort. The Cox regression showed an HR of 0.441 (95% CI, 0.229–0.852) for tumors with high *ZNRF3* expression; furthermore, this factor was positively associated with a better RFS than the loss of its expression.

Ki-67 proliferation marker is considered a prognostic factor in localized ACC and is a powerful tool for predicting recurrence after complete surgical resection [28]. Some studies have documented the importance of Ki-67 as a marker of cellular proliferation, providing a value in the ACC scenario [13,20,91,92]. We categorized the Ki-67 proliferation marker into groups (<10%, ≥10% and <20%, ≥20%), as suggested by Beuschlein et al. [28], and we confirmed the association of Ki-67 with OS and RFS. However, the expressions levels of *ATRX* and *ZNRF3* were not shown to be associated with the Ki-67 proliferation marker.

Since *ATRX* and *ZNRF3* are present in independent pathways, they were considered prognostic markers in this cohort, so we analyzed the association of low expression levels of both. When low expression levels were combined, a stronger negative prognostic value was found for this cohort (HR 0.314; 95% CI, 0.136–0.725; *p* = 0.007).

Tumors with active Wnt/β-catenin signaling due to *CTNNB1*, *ZNRF3*, or *APC* mutations present a dismal prognosis, according to TCGA Research Network. TCGA reported a high expression of *ZNRF3* transcript by RNA sequencing (RNA-seq) levels is associated with a low survival rate in the ACC cohort. On the contrary, we showed that a low *ZNRF3* protein expression is linked to poor survival.

The protein and transcript levels are in agreement in most cases, and some hypotheses may explain the difference in the *ZNRF3* and ACC context. *ZNRF3* is a target gene of Wnt/β-catenin signaling, and the levels of *ZNRF3* transcript can reflect the level of Wnt/β-catenin signaling activity. However, *ZNRF3* deletions may influence its transcript levels, which could be undetectable due to biallelic deletion; simultaneously, the Wnt/β-catenin signaling could be highly active due to the biallelic loss of its *ZNRF3* suppressor. The TCGA analyzed the *ZNRF3* transcript levels without excluding the biallelic deletions cases, and this could be a reason for the difference between transcript and protein expression. Another hypothesis is based on the function of *ZNRF3* negative regulator by R-spondin proteins in the Wnt/β-catenin signaling. *ZNRF3* protein expression may be more dependent on R-spondin proteins activity, through LGR-dependent and LGR-independent mechanisms, reflecting their activity, than the absolute quantity of its transcript.

TCGA also reported *ATRX* transcript expression by RNA-seq levels in the ACC cohort; however, the *ATRX* expression did not impact the survival rate.

Patients classified as having a low or moderate risk of recurrence (ENSAT 1 or 2, presenting microscopically complete resection and Ki-67 ≤10%), according to Fassnacht et al. [8], may benefit from the use of this novel prognostic tool. The presence of low expression levels of *ATRX* and *ZNRF3*, which presented a negative prognostic value, could guide the medical approach used for these particular patients to early adjuvant treatment, since these prognostic markers showed associations with OS and RFS.

Regarding germline *TP53* pathogenic variants, two patients presented *TP53* p.R337H variant (cases 51 and 58), and one patient presented *TP53* p.R273H (case 68) (Appendix A). Although all of them presented with hypercortisolism at initial presentation and died due to ACC (OS 7, 36, and 12 months, respectively), they presented with variable expression of *ATRX* (score 2, 0, and 0, respectively) and variable expression of *ZNRF3* (score 3, 2, and 5, respectively). It remains unclear as to whether an association between the loss of *ATRX* expression and germline *TP53* pathogenic variants exists and could impact OS in the adult ACC population. However, the association between genes has already been published in the pediatric adrenocortical tumor cohort and in other malignancy cohorts [36,57,93,94,95].

Tumor biology knowledge is an important step in the delivery of personalized medicine, allowing the design of better clinical approaches and targeted therapies. This study brought pan-genomic studies closer to clinical practice with an easily accessible tool to refine prognostication in adult patients with ACC. In conclusion, we demonstrated that low levels of protein expression of both *ATRX* and *ZNRF3* are negative prognostic markers of ACC; however, different cohorts should be evaluated in future studies to validate these findings.

## 4. Materials and Methods

The Ethics Committees of the Hospital das Clínicas—University of São Paulo approved this study (n.2.394.934/2017). Informed written consent was obtained from all patients.

All patients underwent an adrenalectomy due to ACC, and they were followed in a unique tertiary center by a team specialized in adrenal disorders. Detailed clinical data, including clinical and hormonal presentations, treatments, follow-up and survival data, imaging, and histopathological results, were collected from medical records. Laboratory results were adjusted according to the delta of reference value, aimed at carrying out a comparison since the laboratory tests were not performed in the same laboratory. ACCs were diagnosed according to Weiss Score (Weiss ≥ 3) [10]. The tumor stage was classified according to the ENSAT and Helsinki systems [5]. Enrolled patients were followed up in the Adrenal Ambulatory-HC/FMUSP and in the Instituto do Cancer do Estado de Sao Paulo—ICESP for an average time of 85.35 months (1.4–406.83 months) (Table 1).

A total of 82 adrenocortical carcinomas were evaluated in terms of the protein expression of *ATRX*, *ZNRF3*, and β-catenin. Detailed information about the cohort is presented in the Appendix A.

ACCs present a large degree of intratumor heterogeneity [9,96], and therefore, two expert adrenal pathologists (I.C.S., M.C.N.Z.) selected the most representative areas of ACC tissues to construct a tissue microarray (TMA). The internal validation of the TMA process has already been published [22,38,97,98,99].

### 4.1. Tissue Microarrays (TMA) and Immunohistochemistry

The tumor series included a total of 82 formalin-fixed paraffin-embedded surgical cases of ACC. Representative areas of the ACCs (viable tumor tissue without necrosis) were identified on hematoxylin- and eosin-stained slides and marked on paraffin donor blocks by two experienced pathologists (I.C.S., M.C.N.Z.). The spotted areas of the donor blocks were punched (1.0 mm punch) and mounted into recipient paraffin blocks using a precision microarray instrument (Beecher Instruments, Sun Prairie, WI, USA), with a total of 3 cores per ACC case. The organization of cores was guided by a cartesian map constituting the three TMAs. Control samples (kidney and liver) were included for TMA orientation. One set of three slides was selected (one slide from each of the three TMA paraffin blocks of the triplicate) for staining with anti-*ATRX* rabbit polyclonal antibody (titer 1:1000; code HPA001906; lot BH115094; 100 μL (0.2 mg/mL); Sigma-Aldrich, Darmstadt, Germany, anti-*ZNRF3* rabbit polyclonal antibody (titer 1:100; code HPA036703; lot A86511; 100 μL (0.1 mg/mL); Sigma-Aldrich, Germany), and anti-β-catenin mouse monoclonal antibody (titer 1:200; clone 14/β-catenin; code 610154; lot 4351813, 150 μg (0.6 mL, 250 μg/mL) BD Transduction Laboratories). The positive control for immunostaining with *ATRX* and *ZNRF3* antibodies was an ACC sample with preserved *ATRX* immunoexpression and with a known wild type *ATRX* status, and a colon carcinoma, respectively. Tonsil tissue was used as a positive control for b-catenin and Ki-67 antibodies. A modified immunoperoxidase immunohistochemical method with humid heat antigen retrieval was used as previously described and validated [38,100]. The immunostaining evaluations were performed by two independent observers (I.C.S. and M.C.N.Z.), who were unaware of clinical data, who independently evaluated *ATRX*, *ZNRF3*, and β-catenin staining. The average value of the two evaluations was taken for statistical analysis [22,97]. TMA samples were included in the analysis only if two or more evaluable cores were available after the staining procedure. Two ACC cases were excluded from analysis due to the loss of tissue sample (1.62%). The interobserver agreement coefficient (kappa) for the staining evaluation was 0.854 (T = 11.76; *p* < 0.0001; Cohen’s kappa coefficient). A kappa coefficient >0.6 is considered to represent substantial agreement [101].

Nuclear staining for Ki-67 was also evaluated, and another set of three slides were stained with mouse monoclonal anti-human Ki-67 antigen (titer 1:40; clone MIB-1; code M7240; lot 95324; 0.2 mL (46 mg/L); Dako, Glostrup, Denmark). The stained slides were scanned by the Scanner of histological slides through Panoramic Viewer 1:15 software (3DHISTECH, Budapest, Hungary). Image Pro Plus 4.5 software (MediaCybernetics, Rockville, MD, USA) was used to process the images, and the positive and total nuclei were automatically counted.

#### Analysis of Immunohistochemistry

*ATRX* and *ZNRF3* expression levels were characterized by nuclear staining and cytoplasmic staining, respectively. Sections were scored semi-quantitatively for extent of *ATRX*, *ZNRF3*, and β-catenin expression in ACC cells as follows: 0, no immunoreactive cells; 1, <25% immunoreactive cells; 2, 25–50% immunoreactive cells; 3, 51–75% immunoreactive cells, and 4, >75% immunoreactive cells. A low expression level of *ATRX* was considered when the presence of <25% immunoreactive cells was identified. The intensity of staining for *ZNRF3* and β-catenin was also scored semi-qualitatively, as follows: 0, negative; 1, weak; 2, intermediate; and 3, strong. The final scores for *ZNRF3* and β-catenin expressions levels were taken as the sum of both parameters (extent and intensity) and grouped into low expression levels (scores of 0–2) and high expression levels (scores of 3–7) as previously reported [99,102]. β-catenin expression in the different cellular localizations (membranous versus cytoplasmic and/or nuclear) was evaluated separately [102,103,104] (Table 2).

### 4.2. Statistical Analysis

The baseline patient characteristics were expressed as absolute and relative frequencies for qualitative variables and as the average, standard deviation, median, minimum, and maximum for quantitative variables. The association among qualitative variables were evaluated by the chi-squared test or Fisher’s exact test. The Spearman’s rank correlation coefficient was calculated for the quantitative variables. The Kaplan–Meier estimator of the survival function was applied for survival analysis, and the log-rank test was used to compare the survival function between groups. Regarding the expression of *ZNRF3* and *ATRX*, the determination of two groups of observations for a simple cut-off point was estimated using the maximum of the standardized log-rank statistic proposed by Lausen and Schumacher in 1992 [105]. The Cox semiparametric proportional hazards model was fitted to describe the relationship between OS or RFS and the covariates [106]. OS was defined as the time from the date of ACC diagnosis to the date of disease-related death or the last follow-up visit. RFS was defined as the time from the date of complete tumor resection to the date of the first radiological evidence of local and/or distant recurrence [38]. The assumption of proportional hazards was assessed based on the so-called Schoenfeld residuals [107,108]. There was evidence that covariates had a constant effect over time in all cases. The significance level was fixed at 5% for all tests. Statistical analyses were performed using IBM SPSS Statistics version 23.0 (IBM Corp., Armonk, NY, USA) and R software version 3.6 (R Foundation for Statistical Computing, Vienna, Austria).

## Figures and Tables

**Figure 1 ijms-22-01238-f001:**
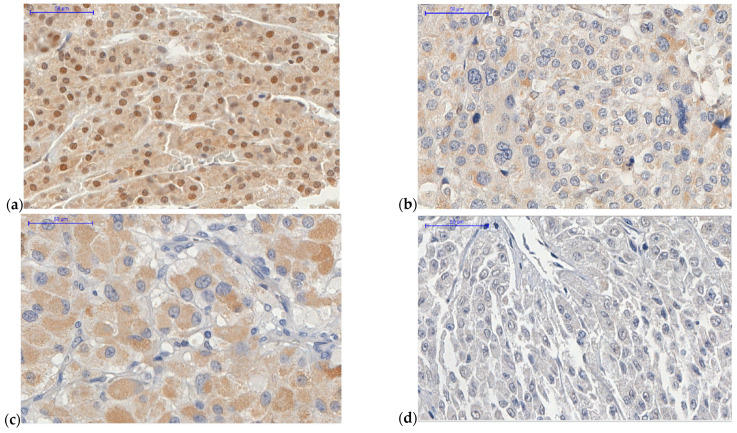
Adrenocortical carcinoma (ACC) samples of tissue microarray (TMA). Obj 40×. (**a**) ACC case 13 presenting high *ATRX* nuclear expression (score 4). Female patient, aged 28.7 y with virilizing syndrome, a Weiss score 3, and a European Network for the Study of Adrenal Tumors (ENSAT) system score of 2. (**b**) ACC case 32 with low *ATRX* nuclear expression (score 0). Female patient, aged 26.5 y with mixed syndrome (hypercortisolism and androgen excess), a Weiss score of 7, and an ENSAT system score of 4. (**c**) ACC case 77 with high *ZNRF3* cytoplasmic expression (score 5; extent = 4, intensity = 1). Male patient, aged 65.6 y with mixed syndrome (hypercortisolism and androgen excess), a Weiss score of 6, and an ENSAT system score of 4. (**d**) ACC case 5 with low *ZNRF3* cytoplasmic expression (score 0). Female patient, aged 85.4 y with hypercortisolism syndrome, a Weiss score of 7, and an ENSAT system score of 3. Scale bar = 50 μm.

**Figure 2 ijms-22-01238-f002:**
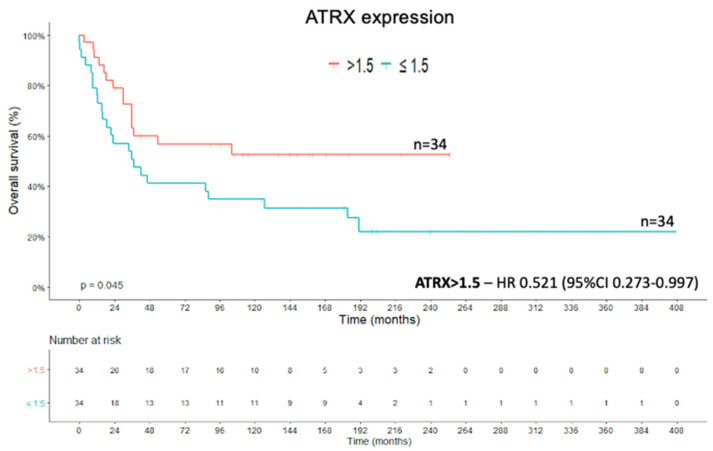
*ATRX* protein expression—overall survival (OS) Kaplan–Meier curve.

**Figure 3 ijms-22-01238-f003:**
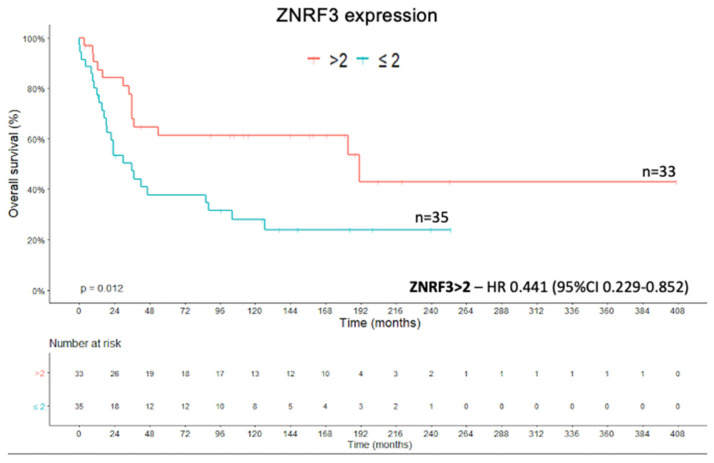
*ZNRF3* protein expression—OS Kaplan–Meier curve.

**Figure 4 ijms-22-01238-f004:**
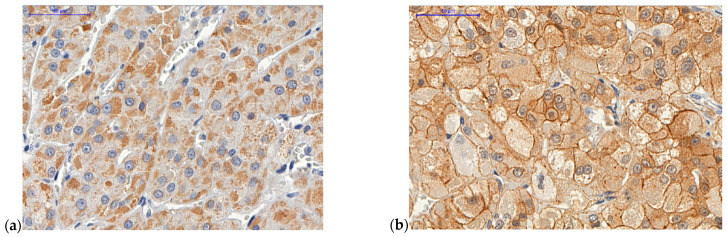
Figure of TMA: (**a**) ACC case 22 with high *ZNRF3* cytoplasmic expression (score 5). Female patient, aged 19.3 y with hypercortisolism syndrome, a Weiss score of 4, and an ENSAT system score of 1. (**b**) ACC case 22 with membranous β-catenin expression. (**c**) ACC case 50 with low *ZNRF3* cytoplasmic expression (score 0). Male patient, aged 36.1y with a silent tumor, a Weiss score of 8, and an ENSAT system score of 2. (**d**) ACC case 50 with cytoplasmic and nuclear β-catenin expression. Scale bar = 50 μm.

**Figure 5 ijms-22-01238-f005:**
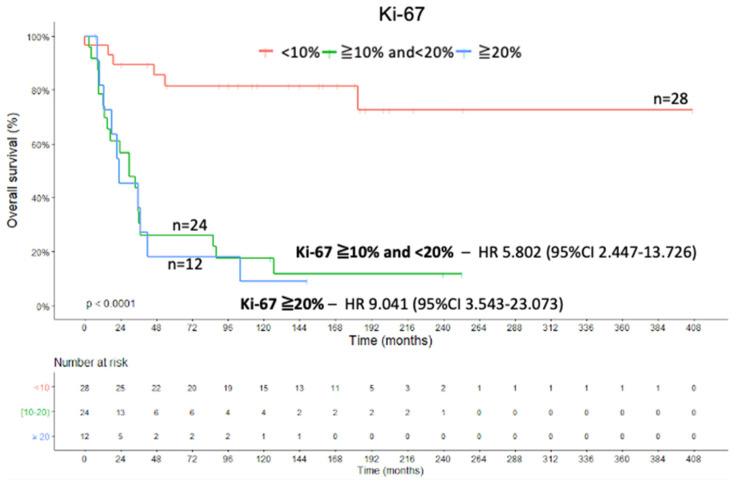
Ki-67 proliferation marker—OS Kaplan–Meier curve.

**Figure 6 ijms-22-01238-f006:**
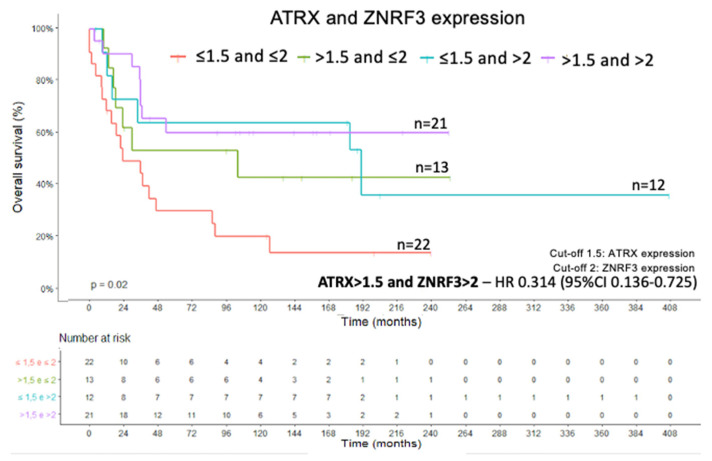
Overall survival curve of combined *ATRX* and *ZNRF3* protein expressions levels.

**Figure 7 ijms-22-01238-f007:**
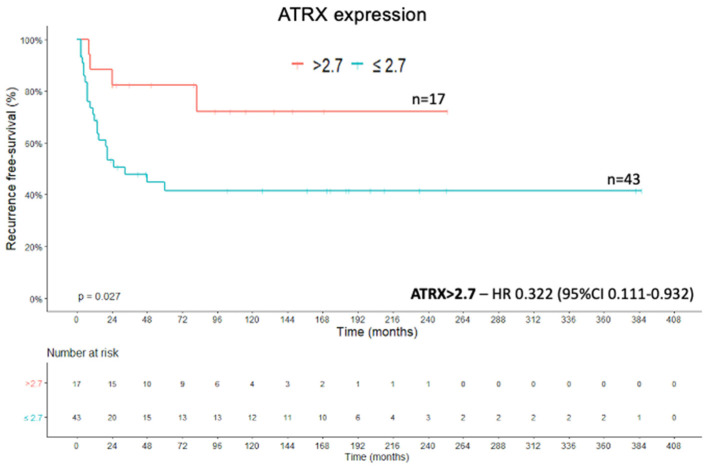
*ATRX* expression—Recurrence-free survival Kaplan–Meier curve.

**Figure 8 ijms-22-01238-f008:**
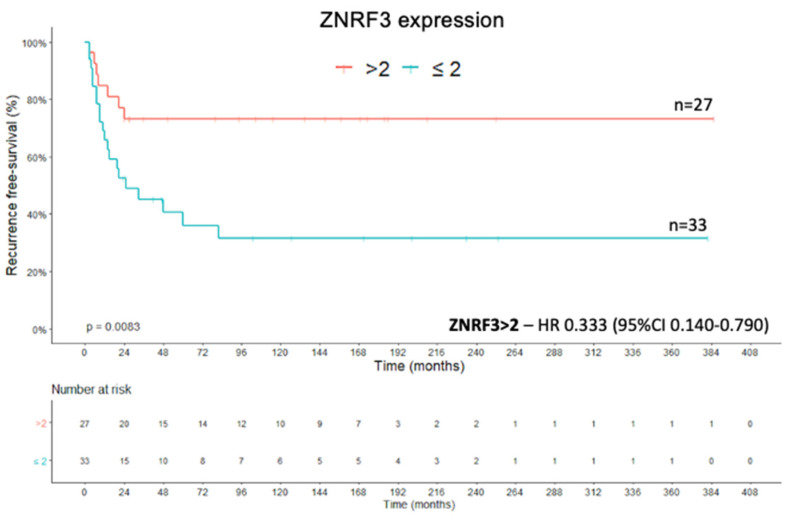
*ZNRF3* expression—Recurrence-free survival Kaplan–Meier curve.

**Figure 9 ijms-22-01238-f009:**
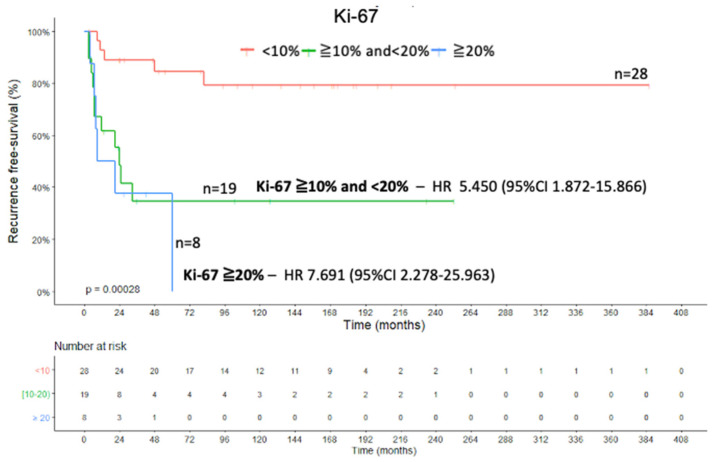
Ki-67 proliferation marker—Recurrence-free survival Kaplan–Meier curve.

**Table 1 ijms-22-01238-t001:** Summary of features of 82 adult patients with adrenocortical carcinoma (ACC) which tumors were used on TMA.

Variables	Parameters	Values—N (%)
**Gender**	Female	63 (76.8%)
	Male	19 (23.2%)
**Age**	Median	38.17 y
	Average	42.05 y
	Range	15.38–85.46 y
**Clinical presentation**	Hypercortisolism	51 (66.2%)
	No hypercortisolism	10 (13%)
	Silent tumor	16 (20.8%)
	Not available	5
**ENSAT * staging system**	1	9 (11.2%)
	2	36 (44.4%)
	3	18 (22.2%)
	4	18 (22.2%)
	Not available	1
**Tumor weight**	Median	445 g
	Average	707.46 g
	Range	10–2600 g
**Tumor size**	Median	11 cm
	Average	11.59 cm
	Range	2.2–23 cm
**Weiss system**	3	15 (19.5%)
	4	16 (20.8%)
	5	6 (7.8%)
	6	12 (15.6%)
	7	11 (14.3%)
	8	15 (19.5%)
	9	2 (2.5%)
**Modified Weiss system**	<3	15 (22.4%)
	3	13 (19.4%)
	4	6 (8.9%)
	5	11 (16.4%)
	6	8 (11.9%)
	7	14 (20.8%)
	Not available	15
**Helsinki system**	≤8.5	21 (32.3%)
	>8.5	44 (67.7%)
	Not available	17
**Ki-67 proliferation marker**	<10%	31 (43%)
	≥10 and <20%	26 (36.1%)
	≥20%	15 (20.9%)
	Not available	10
**Outcome**	Death	39
	Alive	31
	Not available	12
**Global survival**	Median	39.58 months
	Average	85.35 months
	Range	1.4–406.83 months

* ENSAT: The stage classification proposed by the European Network for the Study of Adrenal Tumors.

**Table 2 ijms-22-01238-t002:** Summary of analysis of immunohistochemistry used on TMA for *ATRX* and *ZNRF3* expressions.

Protein Expression	Compartment of ACC Cells	Parameters	Values	Description
***ATRX***	Nuclear	Extent	0	No immunoreactive cells
			1	<25% immunoreactive cells
			2	25–50% immunoreactive cells
			3	51–75% immunoreactive cells
			4	>75% immunoreactive cells
***ZNRF3***	Cytoplasmic	Extent	0	No immunoreactive cells
			1	<25% immunoreactive cells
			2	25–50% immunoreactive cells
			3	51–75% immunoreactive cells
			4	>75% immunoreactive cells
		Intensity	0	Negative
			1	Weak
			2	Intermediate
			3	Strong

## Data Availability

The data presented in this study are available in Appendix A.

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
