# Peer review of "Low Protein Expression of both ATRX and ZNRF3 as Novel Negative Prognostic Markers of Adult Adrenocortical Carcinoma"

_ijms, 2021, doi:10.3390/ijms22031238_

Round 1

Reviewer 1 Report

This is an interesting paper, well written and well design. However some points need to be adressed.

It is a pitty the sample size, and that there os not a validación series. Authors should comment on this un the paper and if posible add more cases ir even better a validación series. Análisis of public databases os Aldo accepted.

Some figures will look better if the HR is added and the groups are grouped together.

Please provide supporting info when analyzing the ihc analysis the way is done.

Confidence intervalo need to be added among the paper. A table with maim resulta of the primary analysis will also be interesting

Author Response

Thank you for the careful review of the manuscript titled Low Protein Expression of both ATRX and ZNRF3 as Novel Negative Prognostic Markers of Adult Adrenocortical Carcinoma.

We are very grateful to the editors and reviewers for the constructive suggestions to improve our paper. We have now incorporated the suggestions proposed by the reviewers. 

You may see more details in the file attached. 

Best Regards. 

Reviewer 2 Report

This work is a report of IHC data on a TMA of Adrenocortical Carcinoma samples. ATRX and ZNRF3 proteins are suggested as a biomarker since low IHC scores of these proteins was correlated with poor survivals. 

In my review, this report cannot be considered for publication on IJMS due to lack of in vitro or in vivo experiments in this work. Also, this report is not in agreement with TCGA dataset which this issue should have been addressed in the discussion properly. 

Author Response

(The authors gave the same response as above.)

Round 2

Reviewer 1 Report

There authors,

Thank you for the replay. It is a pitty that you couldnt use public databases.  With the new changes the work has improved.the abstract needs to be edited and authors should always indicate an statistic like the HR. Moreobrteach pva  new lines introduces on methods could be resume in a new table.

Please  provide a statistic with each pvalue, 

Author Response

Dear Reviewer 1

International Journal of Molecular Sciences,

Thank you for the careful review the manuscript titled Low Protein Expression of both ATRX and ZNRF3 as Novel Negative Prognostic Markers of Adult Adrenocortical Carcinoma.

We are very grateful again for Reviewer 1 for the constructive suggestions to improve our manuscript.

Authors,

Thank you for the replay. It is a pitty that you couldn’t use public databases. With the new changes the work has improved. The abstract needs to be edited and authors should always indicate a statistic like the HR. more breach pva new lines introduces on methods could be resume in a new table.

Please provide a statistic each p value.

We have now incorporated the new suggestions proposed by Reviewer 1. We have edited the abstract, establishing the limit of 200 words (IJMS – Notes for Authors), and adding HR in the results section (Lines 29-42). We created a new table on the section methods (Lines 552 - 554), and we added the statistic value of p-value and a statistic test for all p-values in the manuscript, according to requested.

Sincerely,

Reviewer 2 Report

In this new version of manuscript, a few revisions can be seen while my key concern about TCGA data has not been properly addressed. 

I have attached TCGA data to my review as a PDF file. KM plots shows Low/Medium ZNRF3 samples (n=59) has a poor survival (P=0.05) which is opposite to Fig. 8 of this work. This reviewer is aware of the fact that protein and RNA-seq levels are not essentially in agreement. However, a total different trends should be justified. 

Monir:

1- The emphasis on large cohort is not accurate since the sample size of this work is not really large. 

2- Add sample size to the legends of KM plot for low and high curves.  

Author Response

Dear Reviewer 2

Thank you for the careful review of the manuscript titled Low Protein Expression of both ATRX and ZNRF3 as Novel Negative Prognostic Markers of Adult Adrenocortical Carcinoma.

We are very grateful again for the Reviewer 2 for the constructive suggestions to improve our manuscript.

In this new version of manuscript, a few revisions can be seen while my key concern about TCGA data has not been properly addressed.

I have attached TCGA data to my review as a PDF file. KM plots shows Low/Medium ZNRF3 samples (n=59) has a poor survival (P=0.05) which is opposite to Fig.8 of this work. This reviewer is aware of the fact that protein and RNA-seq level are not essentially in agreement. However, a total different trends should be justified.

Monir:

1- The emphasis on large cohort is not accurate since the sample size of this work is not really large.

2- Add sample size to the legends of KM plot for low and high curves.

We have now incorporated the new suggestions proposed by Reviewer 2. We have clarified the data about recurrence-free survival (Lines 252-254), and we discussed the difference between transcript and protein expressions (Lines 419-435). We deleted the adjective “large” (Lines 31 and 352), and we added the sample sizes of the KM plot (Fig: 2,3,5,6,7,8, and 9).

    Sincerely,